# Abundance of the Dominant Endosymbiont *Rickettsia* and Fitness of the Stored-Product Pest *Liposcelis bostrychophila* (Psocoptera: Liposcelididae)

**DOI:** 10.3390/insects16040349

**Published:** 2025-03-27

**Authors:** Chunqi Bai, Yiwen Duan, Chao Zhao, Lei Yan, Duangsamorn Suthisut, Jianhua Lü, Yueliang Bai, Fangfang Zeng, Meng Zhang

**Affiliations:** 1Henan Collaborative Innovation Center for Grain Storage Security, School of Food and Strategic Reserves, Henan University of Technology, Zhengzhou 450001, China; duanyiwen0704@163.com (Y.D.); zhaochao0310@126.com (C.Z.); hautyanlei@126.com (L.Y.); jianhlv@163.com (J.L.); ylbai@haut.edu.cn (Y.B.); ffzeng1988@163.com (F.Z.); zhangmeng@haut.edu.cn (M.Z.); 2Industry & Technology Innovation Center of Green-Intelligence Control of Stored Products Pests, School of Food and Strategic Reserves, Henan University of Technology, Zhengzhou 450001, China; 3Postharvest Technology on Field Crops Research and Development Group, Postharvest, Processing Research and Development Division, Department of Agriculture, 50 Phaholyothin Road, Ladyao, Chatuchak, Bangkok 10900, Thailand; dsuthisut@yahoo.com

**Keywords:** endosymbiotic bacteria, *Liposcelis bostrychophila*, fitness, stored-product pest

## Abstract

*Liposcelis bostrychophila*, a dominant species of psocids, has become a major pest of stored grain, posing serious threats to grain storage security and safety. Endosymbionts play a significant regulatory role in the population dynamics of their host organisms. This study demonstrated that *Rickettsia* was the predominant genus of symbiotic microorganisms, present in the *Liposcelis bostrychophila* analyzed through metagenomic methods. The abundance of *Rickettsia*, influenced by environmental temperature, positively impacted the fitness of *L. bostrychophila*. Managing the abundance of the endosymbiont *Richettsia* can help control the stored-grain pest *L. bostrychophila.*

## 1. Introduction

*Liposcelis bostrychophila* Badonnel (Psocoptera: Liposcelididae) is a widely distributed pest that infests stored products, including food (wheat, rice, coffee, etc.), medicinal herbs (such as *Corchorus capsularis*), books, archives, and biological specimens [1,2,3]. *L. bostrychophila* can result in a weight loss of 9.7% in broken wheat kernels over three months [4]. Annual losses caused by booklice in commercial rice storage in India (40,000 tons) and Indonesia (150,000 tons) were estimated at GBP 115,000 (USD 180,000) and GBP 50,000 (USD 75,000), respectively [5]. This tiny insect often hides in its habitat, such as grain piles, wall corners, or behind books, and is frequently overlooked; however, heavy infestations can lead to significant economic losses [6]. Additionally, *L. bostrychophila* can carry pathogenic microorganisms and contains allergenic proteins, which may trigger allergic reactions in some individuals [7]. The *Rickettsia felis* strain within *L. bostrychophila* can cause fever and experimental pneumonia in mammals [8]. Additionally, allergenic proteins in booklice are also considered potential factors in triggering allergic reactions. Studies have indicated that the Lip b1 protein in booklice may be a novel allergenic protein, potentially leading to respiratory allergies in humans [9]. Due to its ability to reproduce parthenogenetically, strong mobility, and high adaptability, its population can quickly resurge if control measures are inadequate, making effective pest management a significant challenge [10]. The recovery of booklice populations after chemical pesticide treatment is much faster than that of beetle pest populations [11]. These limitations highlight the need for more innovative and sustainable control methods. Therefore, understanding the factors that influence the population growth of *L. bostrychophila* is essential for successful control efforts.

Insect population growth is influenced by various factors, with endosymbionts playing a significant regulatory role in population dynamics [12]. Endosymbionts are microorganisms that reside within the host’s cells [13], and this symbiotic relationship has existed for approximately 200 to 250 million years [14]. Insect endosymbionts are classified into two categories: obligate endosymbionts and facultative endosymbionts [15,16]. Obligate endosymbionts typically inhabit the host’s reproductive cells or specialized bacteriocytes and are essential for the host’s survival and reproduction, as they are closely linked to the host’s acquisition of vital nutrients [17]. In contrast, facultative endosymbionts are not confined to specific cells [18], and their effects on the host are more varied. These effects can include influencing growth and development [19,20,21], reproduction [22,23], insecticide resistance [24], and heat tolerance [25], etc.

Different types of symbiotic microbes have varying effects on their hosts, such as how these microbes contribute to the host’s nutrition, immunity, and stress tolerance, but also offers new insights into the ecological functions and adaptability of hosts within ecosystems. These symbionts can influence various ecological processes, from nutrient cycling to the resilience of hosts under environmental stress, highlighting their crucial role in the stability and functioning of ecosystems. Earlier research primarily relied on morphological observations, isolation and cultivation techniques, and PCR methods [26,27]. However, The PCR method primarily relies on known sequences, which may not comprehensively capture all species in the host microbiome, especially those that are under-researched or unculturable. Additionally, the quantitative capability of PCR is limited, making it difficult to accurately reflect the richness and diversity of microbial communities. Therefore, with advancements in endosymbiont detection technologies, metagenomic methods have emerged as a unique approach to study the composition and biological functions of insect-associated microbiota [28,29]. For example, the research on the microbial community diversity in the parasitic wasp *Megaphragma amalphitanum* has shown that small body sizes and limited lifespans do not significantly reduce bacterial symbiont diversity [30]. In another study, the bacterial species and their functions in the striped mealybug (*Ferrisia virgata*) were explored, resulting in the successful identification of several bacterial taxa [31]. Additionally, using 16S rRNA gene sequencing, researchers studied the microbial community in the invasive species Asian ladybird, *Harmonia axyridis*, revealing the distribution of dominant bacterial phyla [32]. Similarly, a comparative analysis of the microbiomes of *Diaphorina citri*, *Tamarixia radiata,* and *Diaphorencyrtus aligarhensis*, indicates that *Candidatus* Profftella armature and *Wolbachia* emerge as the dominant endosymbionts [33].

However, studies on the endosymbionts of *L. bostrychophila* have primarily relied on morphological observations or PCR methods that focus on a single species of symbiont bacterium. Yusuf et al. [34] was the first to use electron microscopy to observe a large number of endosymbionts in the reproductive system of *L. bostrychophila*, noting that these endosymbionts were distributed within oocytes. Based on their morphological characteristics, he identified the symbiont as being similar to *Wolbachia*, a finding that was later confirmed through 16S rDNA amplification. Additionally, treatment with 1% rifampicin resulted in the elimination of this symbiont, leading to a significant decrease in egg-laying by *L. bostrychophila* [35]. Mikac [36] employed *Wolbachia*-specific primers for PCR amplification and successfully detected the presence of *Wolbachia*. Wang [37] studied three geographical populations of *L. bostrychophila* using 16S rRNA universal primers, identifying a symbiotic bacterial fragment of approximately 1500 bp. Phylogenetic analysis indicated that this bacterium is closely related to *Rickettsia* in the class Alpha-proteobacteria and is also genetically similar to *W. pipientis*, although its exact species could not be determined. Wang et al. [38] used *Cardinium*-specific primers for PCR amplification and confirmed the presence of *Cardinium* in *L. bostrychophila*. Following four weeks of treatment with 1% rifampicin, *Cardinium* was completely eliminated, resulting in a significant reduction in the insect’s fitness. Meanwhile, Perotti et al. [39] utilized both specific and universal primers for PCR amplification; however, only *Rickettsia* was amplified, with no detection of *Wolbachia* or *Cardinium*. After subjecting the samples to heat treatment at 37 °C to eliminate *Rickettsia*, egg-laying in *L. bostrychophila* decreased by approximately 90%. Behar et al. [40] used specific primers for 16S rRNA, *gltA*, *ompA*, *ompB*, and the cell surface antigen *sca4* to conduct amplification and sequencing, confirming that the endosymbiont in *L. bostrychophila* is *R. felis*. To date, research on the endosymbionts of *L. bostrychophila* has primarily focused on these three types of symbionts.

However, the overall diversity of its endosymbionts, the dominant symbiont species, and their relationship with host fitness remain largely unexamined. Therefore, we detected the diversity of symbiotic bacteria in *L. bostrychophila* through metagenomic sequencing and identified the dominant endosymbiotic bacteria species in it. By treating the insects at different temperatures, we established experimental systems of *L. bostrychophila* with different abundances of dominant endosymbiotic bacteria. Using the life table method, we counted the developmental duration, survival rate, fecundity, etc., of each life stage of *L. bostrychophila* under different abundances of dominant endosymbiotic bacteria. In-depth study of the abundance of dominant endosymbiotic bacteria and their impact on the fitness of *L. bostrychophila* will provide key clues for understanding the ecological adaptation mechanism of this pest. This will not only help to reveal the role of endosymbiotic bacteria in insect population dynamics, but may also provide potential biological control methods for future pest management strategies.

## 2. Materials and Methods

### 2.1. Insect

*Liposcelis bostrychophila* specimens were collected from a grain reserve warehouse in Wuhan, Hubei Province, China. They were subsequently purified and cultured for several generations in the Stored-Product Pest Control Laboratory at Henan University of Technology (HAUT). Approximately 500 adult *L. bostrychophila* were placed in clean, disposable plastic rearing dishes (diameter = 4.5 cm, height = 2.5 cm), with about 25 individuals per dish, using a total of 20 rearing dishes. Appropriate feed, consisting of wheat flour, skim milk powder, and yeast powder in a mass ratio of 10:1:1, was added. The insects were cultured in the dark at a temperature of (28 ± 1) °C and (75 ± 2.5) % relative humidity (RH). After three days, all adults were removed, and three-day-old eggs were collected. The eggs continued to be cultured under the same conditions for 30 days; after hatching, nymphs were cleaned from any attached feed and impurities using a crawler tower. The feed consisted of a mixture of whole wheat flour (screened through a 100-mesh sieve), yeast powder, and skimmed milk powder in a 10:1:1 ratio. The crawler tower was constructed using two clean glass culture dishes: the lower dish (diameter = 9 cm, height = 2 cm) served as a lid, with the opening facing upward, while the upper dish (diameter = 3.5 cm, height = 1.5 cm) was inverted and placed on top of the lower dish. The experimental insects were placed in the upper part of the crawler tower, allowing them to move freely. Those individuals collected from the lower part were carefully removed under a microscope using a fine brush to obtain clean experimental insects.

### 2.2. Metagenomic Sequencing

Clean *L. bostrychophila* adults were placed into sterilized centrifuge tubes, with approximately 5000 individuals per tube, filling each tube up to 1.5 mL. Under sterile conditions, the insects were washed three times with deionized distilled water (ddH_2_O) and 75% ethanol to ensure surface sterilization. The samples were labeled as HBSJ1. Similarly, additional samples, labeled as HBSJ2 and HBSJ3, were prepared. All samples were stored at −80 °C. They were then sent to Shanghai Shenggong Biotechnology Co., Ltd. (Shanghai, China) for metagenomic sequencing. The specific steps are as follows:

#### 2.2.1. DNA Extraction

Total community genomic DNA extraction was performed using a E.Z.N.A. Soil DNA Kit (Omega, M5635-02, Norcross, GA, USA), following the manufacturer’s instructions. We measured the concentration of the DNA using a Qubit 4.0 (Thermo, Waltham, MA, USA) to ensure that adequate amounts of high-quality genomic DNA had been extracted.

#### 2.2.2. Library Preparation for Sequencing

The total amount of DNA in each sample was 500 ng, as input material for DNA sample preparation. Sequencing libraries were generated using Hieff NGS^®^ MaxUp ||DNA Library Prep Kit for Illumina^®^ (12200ES96, YEASEN, Shanghai, China) following manufacturer’s instructions, and index codes were added to attribute sequences to each sample. In a nutshell, DNA was broken into fragments of about 500 bp using Covaris 220. In order to purify DNA fragments of 500 bp, the library fragments were purified with Hieff NGS™ DNA Selection Beads (12601ES56, YEASEN, China). The purification is considered successful when the total amount of the purified product reaches 40% of the initial input. The purified DNA was end-repaired and Adapter-Ligated, and underwent fragment selection. Then, PCR was performed with 2 × Super Canace^®^II High-Fidelity Mix, Primer Mix(p5/p7) and Adaptor-Ligated DNA. At last, PCR products were purified (Hieff NGS™ DNA Selection Beads) and library quality was assessed on the Qubit^®^4.0 Flurometer. The libraries were then quantified and pooled. Paired-end sequencing of the library was performed on the NovaSeq 6000 sequencers (Illumina, San Diego, CA, USA).

#### 2.2.3. Data Assessment and Quality Control

Fastp (version 0.36) was used for evaluating the quality of sequenced data. Raw reads were filtered according to several steps: (1) removing adaptor sequence; (2) removing low quality bases from reads 3′ to 5′ (Q < 20), using a sliding window method to remove base values less than 20 in tail reads (window size is 4 bp); (3) finding the overlap of each pair of reads and properly correcting inconsistent bases within the interval; (4) removing reads with read lengths of less than 35nt and their pairing reads. And the remaining clean data were used for further analysis.

#### 2.2.4. Metagenome Assembly and Binning

First, use Megahit (version 1.2.9) to perform multi-sample mixed splicing to obtain preliminary spliced contig sequences. Then, use bowite2 (version 2.1.0) to map clean reads were back to the spliced results, extract unmapped reads, and splice them again using SPAdes (version 3.13) to obtain low-abundance contigs. The parameters for Bowtie2 are as follows: D 20 -R 3 -N 1 -L 20 -i S,1,0.50. MetaWRAP (version 1.3.2) was used to perform a binning series, and processes such as Bin sorting, Bin purification, Bin quantification, Bin reassembly, and Bin identification were performed in sequence. After filtering, a draft genome of a single bacteria with high integrity and low contamination was obtained.

#### 2.2.5. Gene Prediction and Non-Redundant Gene Set Construction

Prodigal (version 2.60) was used to predict the ORF of the splicing results, select genes with a length greater than or equal to 100 bp, and translate them into amino acid sequences. For the gene prediction results of each sample, the CD-HIT (version 2.60) was used for de-redundancy, to obtain a non-redundant gene set. Salmon (version 1.5.0) was used to construct a specific index of non-redundant gene sets, using a dualphase algorithm and a method of constructing a bias model to accurately quantify the abundance of genes in each sample, and calculate gene abundance based on gene length information.

#### 2.2.6. Species Annotations

DIAMOND (version 0.8.20) was used to compare the gene set with NR to obtain species annotation information. Screening conditions: E-value < 1 × 10^−5^, Score > 60. Based on gene set abundance information and annotation information, species abundances were obtained.

### 2.3. Construction of Diverse Rickettsia Abundance System

Two thousand *L. bostrychophil* adults were placed in several rearing boxes and provided with appropriate feed. The insects were cultured in the dark at a temperature of (28 ± 1) °C and (75 ± 2.5) % RH. After 24 h, the adults were removed, and 1-day-old eggs (*E*0) were collected. The *E*0 were divided into three groups: *a*0, *a*1, and *a*2. Each group was placed into separate rearing boxes under the following conditions: the *a*0 group was cultured at (28 ± 1) °C, the *a*1 group at (35 ± 1) °C, and the *a*2 group at (37 ± 1) °C. All groups were maintained at (75 ± 2.5) % RH in dark conditions. This setup aimed to establish high, medium, and low abundance populations of Rickettsia in *L. bostrychophila*, referred to as *LBhp*, *LBmp*, and *LBlp*, respectively. Samples were collected at various developmental stages: 1-day-old eggs, first nymphs, second nymphs, third nymphs, fourth nymphs, and adults. For instance, at the first nymph stage of *LBhp*, 50 nymphs of similar size were randomly selected as a sample using the crawler tower method. Three samples were collected in total. The insects were washed three times with ddH_2_O and 75% ethanol. DNA was then extracted and subjected to quantitative PCR analysis. Details of the quantitative fluorescence PCR analysis method and specific information about the primers are provided in the Appendix A. The abundance of *Rickettsia* was assessed using a standard curve. The same method was applied to the other developmental stages to evaluate the abundance differences of in *LBhp*, *LBmp* and *LBlp Rickettsia*.

### 2.4. Fitness of L. bostrychophila at Varying Levels of Rickettsia Abundance

Five hundred adults of *L. bostrychophila* were placed in insect rearing boxes, and an appropriate amount of feed was added. After incubating for 24 h at (28 ± 1) °C and (75 ± 2.5) % RH, in dark conditions, the adults were removed to obtain 1-day-old eggs (*E*0). The *E*0 eggs (150 eggs) were evenly divided into three groups: *g*0, *g*1 and *g*2. The rearing conditions for *g*0, *g*1, and *g*2 corresponded to the conditions for *LBhp*, *LBmp* and *LBlp*, respectively. Fitness parameters of *L. bostrychophila* were recorded, and developmental durations, survival rates, lifespans, reproductive periods, and reproductive abilities at each stage were analyzed using the TWOSEX-MSChart software (Version 5/7, 2024) [41]. Population growth parameters, including generation time (*T*), net reproductive rate (*R*_0_), intrinsic rate of increase (*r*), and finite rate of increase (*λ*), were calculated [42,43,44,45].

### 2.5. Data Statistics and Analysis

The age–stage specific survival rate (*S*_*x**j*_), female adult age-stage reproductive value (*f_xj_*), population age–stage specific survival rate (*l*_*x*_), population age-specific fecundity (*m*_*x*_), population age-specific net reproductive rate (*l*_*x*_*m*_*x*_), and specific age–stage reproductive value (*v*_*x**j*_) were derived using the TWOSEX-MS Chart software and plotted using Origin 2021 software. The mean values and standard errors of life table parameters were calculated using the bootstrap method. The Paired Bootstrap Test (TWOSEX-MS Chart) program was used to estimate the significance (*p* < 0.05) of differences in population growth parameters and the fecundity of *L. bostrychophila* with different *Rickettsia* abundances [41,44]. One-way analysis of variance (ANOVA) was performed using IBM SPSS 27 software to analyze the differences in developmental time, nymphal stage duration, longevity, pre-oviposition period, oviposition period, and fecundity. Duncan’s multiple range test (*p* < 0.05) was used to assess statistical differences between means.

## 3. Results

### 3.1. Dominant Symbiotic Bacterial Species in L. bostrychophila

Based on the metagenomic sequencing results of the symbiotic bacteria in *L. bostrychophila*, a total of 51 genera were identified, with the dominant genus being *Rickettsia* (84.11–98.16%). At the species level, the dominant species was *R. felis* (66.17–78.45%), as shown in Figure 1.

### 3.2. Quantification of Rickettsia

The melt curve displayed a clear single peak, as shown in Figure 2, with no non-specific amplification products or primer dimers, indicating that the *Rickettsia* genus-specific target gene *Rb* amplification product was specific, and that the primers were suitable for quantitative analysis of the *Rb* gene. The standard curve obtained for the plasmid was Y = −2.968x + 36.14, with a correlation coefficient of R^2^ = 0.999, as shown in Figure 3, indicating a good linear relationship between the Ct values and the log values of plasmid copy numbers.

### 3.3. Diverse Rickettsia Abundance System

The adult insects observed from eclosion to one-week post-eclosion are illustrated in Figure 4. Significant differences in *Rickettsia* abundance were noted among the *a*0, *a*1, and *a*2 populations, demonstrating the successful establishment of a differential system for the dominant symbiotic bacterium *Rickettsia* in *L. bostrychophila*. The *a*0 population, cultured at a temperature of (28 ± 1) °C, showed the highest *Rickettsia* abundance, categorizing it as *LBhp*. The *a*1 population, maintained at (35 ± 1) °C, had a medium level of *Rickettsia* abundance; thus, it is referred to as *LBmp*. The *a*2 population, cultured at (37 ± 1) °C, exhibited the lowest *Rickettsia* abundance, and is classified as *LBlp*. Although we conducted experiments under different temperature conditions, we did not observe the complete disappearance of *Rickettsia* at any temperature. No significant differences in *Rickettsia* abundance were observed among the *a*0, *a*1, and *a*2 populations across early developmental stages (egg, first nymph, and second nymph). However, significant differences were found between the *a*0 and *a*2 populations at the third nymph stage, and between the *a*0 population and both the *a*1 and *a*2 populations at the adult stage.

### 3.4. Life Table Parameters of L. bostrychophila at Varying Levels of Rickettsia Abundance

The egg hatching times of *L. bostrychophila* in *LBhp* (*g*0 group), *LBmp* (*g*1 group), and *LBlp* (*g*2 group) were significantly different (see Table 1). Compared to *LBhp*, the hatching times were significantly shorter in the *LBmp* and *LBlp* groups, with hatching durations of 7.98 days, 5.64 days, and 6.87 days, respectively. In terms of nymph development, both the *LBmp* and *LBlp* groups had three instars, while the *LBhp* group had four instars. As the abundance of the dominant symbiotic bacteria decreased, the lifespan of *L. bostrychophila* was significantly reduced, resulting in lifespans of 217.4 days for *LBhp*, 121.98 days for *LBmp*, and 63.88 days for *LBlp*. Additionally, fecundity significantly declined, with egg production recorded at 155.56, 72.71, and 0.00 eggs for the *LBhp*, *LBmp*, and *LBlp* groups, respectively. No males were recorded at any of the experimental temperatures.

### 3.5. Population Parameters of L. bostrychophila at Varying Levels of Rickettsia Abundance

The population parameters were analyzed using the bootstrap method. The results indicated that the generation time (*T*), net reproductive rate (*R_0_*), intrinsic growth rate (*r*), finite rate of increase (*λ*), and fecundity of *LBhp* were significantly higher than those of *LBmp* and *LBlp*. The *LBlp* exhibited an inability to lay eggs normally, as shown in Table 2.

### 3.6. S_xj_ of L. bostrychophila at Varying Levels of Rickettsia Abundance

The *S*_*x**j*_ of *L. bostrychophila* populations with varying *Rickettsia* abundances are illustrated in Figure 5. The data indicate that Rickettsia abundance significantly affects the survival rate of *L. bostrychophila*. Specifically, compared to *LBhp*, the survival rate of adult *L. bostrychophila* was markedly lower in *LBmp* and *LBlp*. Among the different populations, *LBhp* exhibited the highest peak survival rate and the longest lifespan, followed by *LBmp*. In contrast, *LBlp* had the lowest peak survival rate and the shortest lifespan. Overall, a decrease in *Rickettsia* abundance correlated with a decline in the lifespan of *L. bostrychophila*.

### 3.7. l_x_ and mx of L. bostrychophila at Varying Levels of Rickettsia Abundance

The *l*_*x*_ of *L. bostrychophila* in *LBhp* began to decline from 120 days, and as the age increased, *l*_*x*_ gradually decreased from 98% to 0%, as shown in Figure 6a. In *LBmp*, *l*_*x*_ started to decline from 10 days, dropping to 90% by 11 days. After that, *l*_*x*_ value remained relatively stable between 11 days and 81 days, then gradually decreased from 90% to 0%, as shown in Figure 6b. In *LBlp*, *l*_*x*_ started to decline from day 10, reaching 66% by 13 days. After that, *l*_*x*_ remained stable between 13 days and 51 days, then gradually decreased from 66% to 0%, as shown in Figure 6c. The *f_xj_* of female *L. bostrychophila* in both *LBhp* and *LBmp* showed a trend of increasing and then decreasing. However, the peak of the fecundity in *LBmp* was higher than in *LBhp*, and the duration was shorter. The *LBlp* was unable to lay eggs, and its fecundity remained at 0.

The *v*_*x**j*_ curve of *L. bostrychophila* in *LBhp* exhibited periodic fluctuations, with a peak value of 17.61 occurring at 23 days, as shown in Figure 7a. The *v*_*x**j*_ curve of *L. bostrychophila* in *LBmp* was unimodal, with a peak value of 20.40 occurring at 17 days, as shown in Figure 7b. Although the peak reproductive value in *LBmp* was higher and occurred earlier, it rapidly declined to 0 after the peak. In contrast, the reproductive value in *LBhp* decreased more slowly after reaching its peak, maintaining a value of 12.61 on 78 days.

## 4. Discussion

Regarding the role of *Rickettsia* in *L. bostrychophila*, the metagenomic results revealed that the symbiotic microbiome of *L. bostrychophila* collected from Hubei is complex and composed of various microorganisms. This differs from previous studies using PCR or RT-PCR techniques, which showed that *L. bostrychophila* would be infected with *Rickettsia*, *Wolbachia*, or *Cardinium* [35,36,38,39]. The results indicate that *Rickettsia* was the dominant genus in the symbiotic microbiome of *L. bostrychophila*, consistent with the findings of Perotti [39], with the dominant species being *R. feli*.

As for the impact of *Rickettsia* on the biological characteristics of *L. bostrychophila*, by controlling the growth temperature of *L. bostrychophila* at 28 °C, 35 °C, and 37 °C, significant differences in the abundance of *Rickettsia* infection were observed. This aligns with findings from studies on *L. tricolor* [46], providing valuable insights for future research on the relationship between the abundance of different symbiotic bacteria and their hosts. Furthermore, since the temperature range examined in this study is similar to the environmental conditions where *L. bostrychophila* is found in the field—specifically, warm and humid environments, with a preference for temperatures between 20 °C and 42 °C [47]—this research has potential practical applications and offers important reference values.

In this study, we found that the fitness of *L. bostrychophila* is closely linked to the abundance of the symbiotic bacterium *Rickettsia*. Changes in *Rickettsia* abundance significantly affected several biological indicators of *L. bostrychophila*, including egg production, survival rate, and longevity. Specifically, we observed that egg production in *L. bostrychophila* was notably reduced when *Rickettsia* abundance decreased. Similarly, Yusuf’s research indicated that egg production in *L. bostrychophila* significantly declined following the removal of *Rickettsia* [34]. Perotti’s study found a dramatic decrease of approximately 90% in egg production after *Rickettsia* was eliminated through heat treatment at 37 °C [39]. However, our results showed that at 37 °C, *L. bostrychophila* was unable to lay eggs after developing from egg to adult, which differs from Perotti’s findings.

The reduction in *Rickettsia* abundance significantly impacted the age–stage specific survival rate (*S*_*x**j*_), which accurately reflects the survival and aging differentiation of *L. bostrychophila* populations. Our study revealed that nymph survival peaked in both *LBhp* and *LBmp*, and was significantly higher than in *LBlp*. Furthermore, both the peak adult survival rate and its duration were much longer in *LBhp* compared to *LBmp* and *LBlp*. Additionally, the longevity of L. bostrychophila decreased as *Rickettsia* abundance diminished. These results are similar to those of a study on the effects of the endosymbiont *Rickettsia* on the biological characteristics of the whitefly, where *Rickettsia* infection was found to be able to extend the lifespan of adult whiteflies [48]. Overall, the reduction in *Rickettsia* abundance considerably lowered the fitness of *L. bostrychophila*. In terms of *Rickettsia’*s effects on other hosts, some research has shown that *Rickettsia* infection can enhance the survival and reproductive capacity of the whitefly, *Bemisia tabaci* [49], which aligns with our findings. However, *Rickettsia* did not significantly affect the fitness of the sweet potato whitefly, *B. tabaci* [50]. Moreover, some studies indicated that *Rickettsia* had minimal impact on the embryo development and egg-laying of *Ixodes pacificus* ticks [51]. Chen’s research also found that *Rickettsia* led to a decrease in the reproductive capacity of pea aphids, *Acythosiphon pisum* [21]. These findings underscore that the impact of *Rickettsia* on the fitness of different insect species varies.

In this study, we did not specifically assess the potential role of *Rickettsia* as a reproductive control factor. However, previous studies have shown that *Rickettsia* can induce changes in reproductive patterns in some species. For example, *Rickettsia* can cause parthenogenesis in the parasitoid wasp *Neochrysocharis formosa* and the parasitic wasp *Pnigalio soemius* [52]. These findings suggest that *Rickettsia* may have reproductive control potential in some insect species. Whether *Rickettsia* induces parthenogenesis in the *L. bostrychophila* is worth exploring further in future studies.

Our study results indicate that *Rickettsia* was the dominant endosymbiont in *L. bostrychophila* and had a significant impact on its fitness. However, the diversity of symbiotic microorganisms in insects is influenced by various factors, and the diversity and proportions of these microorganisms may vary among different geographic populations. Additionally, symbiotic bacteria in insects often collaborate to perform specific functions, such as cellulose degradation or the synthesis of essential amino acids [53]. It is important to note that this study focused on a single geographic strain of *L. bostrychophila*, leaving us uncertain about whether the diversity of symbionts and their associated functions are consistent across other populations. Future research should aim to collect additional geographic strains of *L. bostrychophila* for more comprehensive investigation. In our research, we observed that differences in endosymbiont abundance were influenced by developmental temperatures following egg laying, which may be related to temperature variations. Therefore, the research system needs to be optimized to better understand the relationship between *Rickettsia* and *L. bostrychophila*.

## 5. Conclusions

*Rickettsia* serves as a secondary endosymbiont in L. bostrychophila and plays a crucial role in its biological processes. In this study, we utilized metagenomic techniques to investigate the diversity of symbiotic bacteria present in *L. bostrychophila*. Our results indicated that *Rickettsi*a has the highest relative abundance and is identified as the dominant endosymbiont in this organism. We created populations of *L. bostrychophila* with varying levels of *Rickettsia* abundance by controlling the temperature and further examined how these varying levels affected the fitness of *L. bostrychophila*. The findings revealed that as the abundance of *Rickettsia* decreased, the fitness of *L. bostrychophila* significantly declined. However, due to the study’s limitations, further in-depth research is necessary to gain a better understanding of *Rickettsia*’s specific effects on egg-laying and development in *L. bostrychophila*, as well as the underlying mechanisms involved.

## Figures and Tables

**Figure 1 insects-16-00349-f001:**
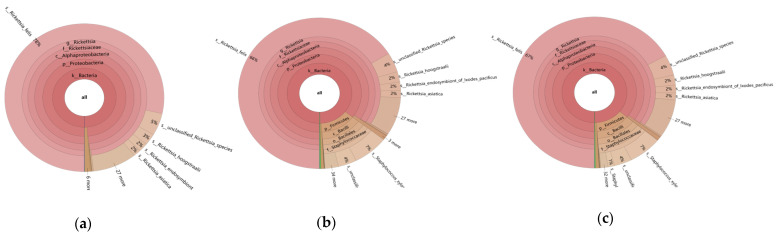
Multilevel species composition chart of endosymbiotic bacteria of *L. bostrychophila.* (**a**) HBSJ1; (**b**) HBSJ2; (**c**) HBSJ3.

**Figure 2 insects-16-00349-f002:**
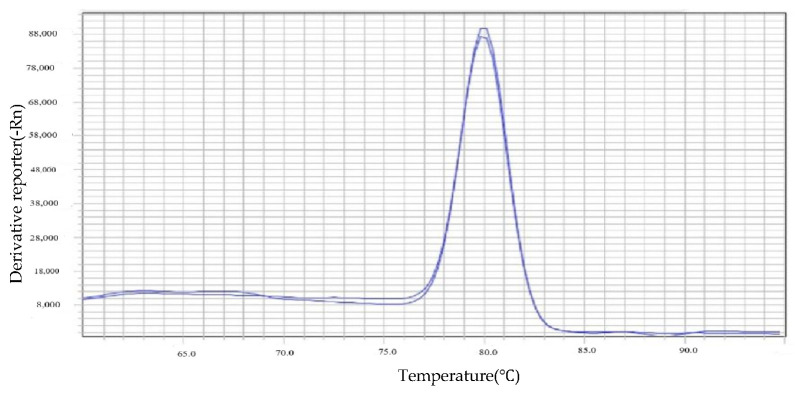
*Rb* gene dissolution curve.

**Figure 3 insects-16-00349-f003:**
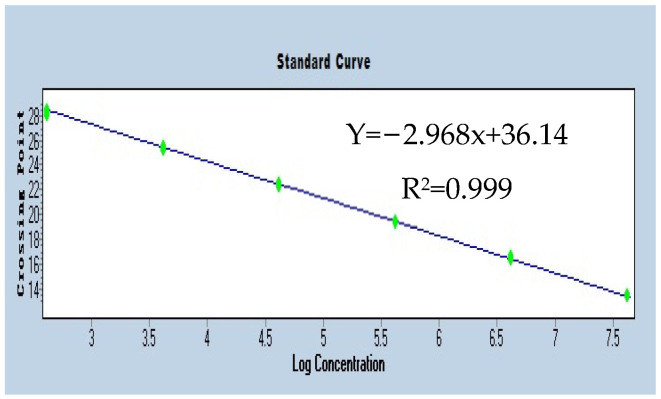
*Rb* gene qPCR standard curve.

**Figure 4 insects-16-00349-f004:**
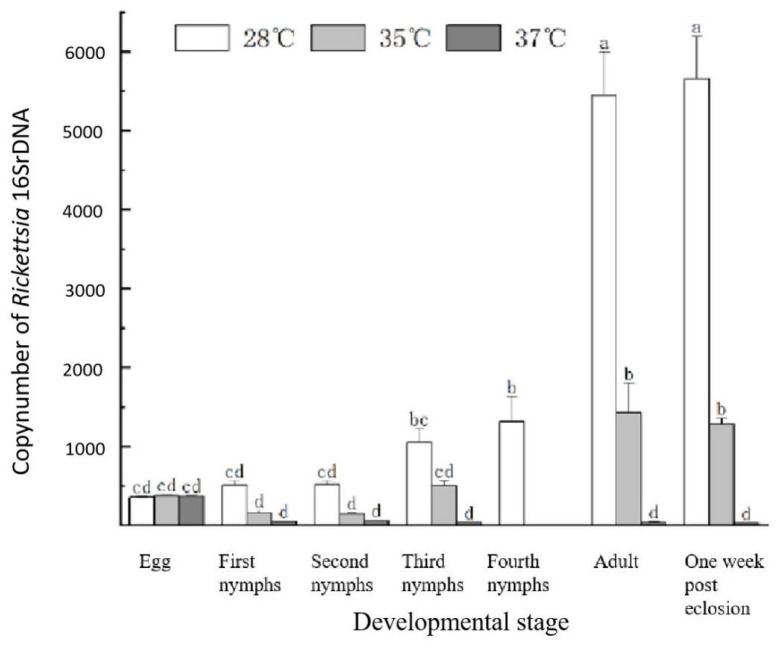
Copy number of *Rickettsia* in *L. bostrychophila* at different temperatures. Scale bars with different letters are significantly different (*p* < 0.05).

**Figure 5 insects-16-00349-f005:**
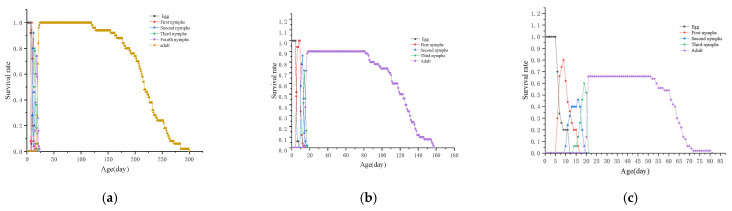
*S**x**j* curve of *L. bostrychophila* at varying levels of *Rickettsia* abundance. (**a**) *LBhp*; (**b**) *LBmp*; (**c**) *LBlp*.

**Figure 6 insects-16-00349-f006:**
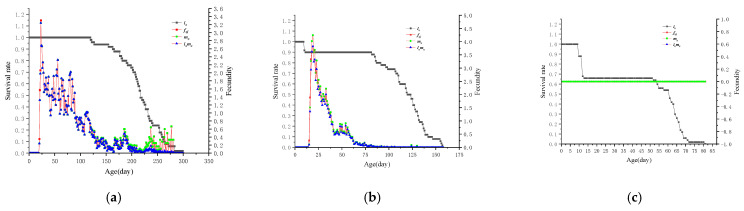
The population age–stage specific survival rate (*l*_*x*_), female adult age–stage specific fecundity (*f*_xj_), population age-specific fecundity (*m*_*x*_), and population age-specific net reproductive rates (*l*_*x*_*m*_*x*_) of *L. bostrychophila* at varying levels of *Rickettsia* abundance. (**a**) *LBhp*; (**b**) *LBmp*; (**c**) *LBlp*.

**Figure 7 insects-16-00349-f007:**
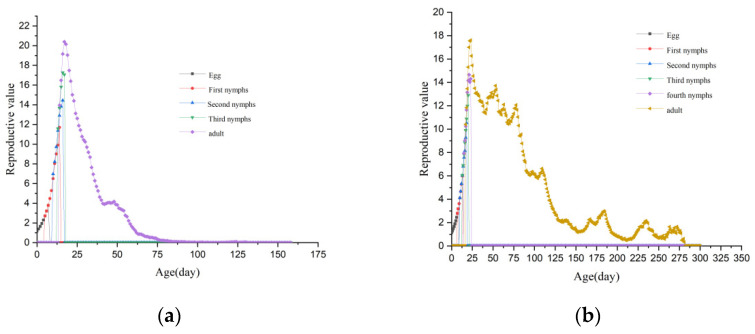
The age–stage specific reproductive value (*v*_*x**j*_) of *L. bostrychophila* at varying levels of *Rickettsia* abundance. (**a**) *LBhp*; (**b**) *LBmp*.

**Table 1 insects-16-00349-t001:** Life table parameters of *L. bostrychophila* at varying levels of *Rickettsia* abundance.

Levels of *Rickettsia* Abundance	Developmental Periods (d)						Longevity (d)	Adult Preoviposition Period (d)	Oviposition Period (d)	Fecundity (Eggs/Female)
	Egg (d)	First Nymphs (d)	Second Nymphs (d)	Third Nymphs (d)	Fourth Nymphs (d)	Nymph Period (d)				
*LBhp*	7.98 ± 0.05 a	3.06 ± 0.13 a	2.78 ± 0.16 a	3.42 ± 0.22 a	2.94 ± 0.15 a	12.2 ± 0.17 a	217.4 ± 5.70 a	1.54 ± 0.12 a	176.66 ± 5.59 a	155.56 ± 3.87 a
*LBmp*	5.64 ± 0.11 b	4.9 ± 0.18 b	3.22 ± 0.14 a	1.98 ± 0.12 b	—	10.11 ± 0.19 b	121.98 ± 3.00 b	0.84 ± 0.14 b	55.42 ± 2.87 b	72.71 ± 2.06 b
*LBlp*	6.87 ± 0.14 c	5.8 ± 0.38 c	4.45 ± 0.35 b	3.18 ± 0.29 a	—	13.88 ± 0.14 c	63.88 ± 1.03 c	0.00 ± 0.00 c	0.00 ± 0.00 c	0.00 ± 0.00 c

Note: The data in the table are expressed as mean ± standard error. Different letters within the same row indicate a significant difference (*p* < 0.05). In this table, all instances of “d” represent “days”.

**Table 2 insects-16-00349-t002:** The population parameters of *L.bostrychophila* at varying levels of *Rickettsia* abundance.

Parameter	Levels of *Rickettsia* Abundance		
	*LBhp*	*LBmp*	*LBlp*
*T*	39.27 ± 0.50 a	25.11 ± 0.21 b	—
*R_0_*	155.56 ± 3.84 a	65.44 ± 3.59 b	—
*r*	0.13 ± 0.002 b	0.17 ± 0.003 a	—
*λ*	1.14 ± 0.002 b	1.18 ± 0.003 a	—

Note: The data in the table are expressed as mean ± standard error. Different letters within the same row indicate a significant difference (*p* < 0.05).

## Data Availability

The original contributions presented in this study are included in the article and Appendix A; further inquiries can be directed to the corresponding authors.

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
