# Peer review of "Abundance of the Dominant Endosymbiont Rickettsia and Fitness of the Stored-Product Pest Liposcelis bostrychophila (Psocoptera: Liposcelididae)"

_insects, 2025, doi:10.3390/insects16040349_

Round 1
Reviewer 1 Report
Comments and Suggestions for Authors
Comments and Suggestions for Authors
I have read with attention the ms titled: "Effect of the Abundance of the Dominant Endosymbiont Rickettsia on the Fitness of the Stored-Product Pest Liposcelis bostrychophila (Psocoptera: Liposcelididae)” and authored Bai Chunqi, Duan Yiwen, Zhao Chao, Yan Lei, Duangsamorn Suthisut, LV Jianhua, Bai Yueliang, Zeng Fangfang and Zhang Meng
Below you will find my remarks, comments, and suggestions.
Lines 43–46: Can authors provide some values for the economic losses caused by L. bostrychophila globally or regionally? This data would be invaluable in highlighting the pest's importance to the general audience. Additionally, given the authors' expertise, could they provide some examples of stored products, such as what kind of food or medicinal herbs, to further enrich the manuscript?
Lines 47–49: Could the authors kindly expand on the nature of pathogenic microorganisms and allergenic proteins carried by L. bostrychophila? I understand that the authors' knowledge in this area is extensive, and I believe that providing specific diseases or allergens would further strengthen the argument.
Lines 50–52: it would be a good idea to mention why current control measures fail or their limitations in addressing these traits. This will provide a seamless transition into the study's significance.
Lines 54–64: It is a well-established fact that there is a link between endosymbionts and population growth. It is important, but if the authors can provide a brief explanation of why this is relevant to L. bostrychophila, it would be a great addition. It would also be great if the authors could explain how endosymbionts influence pest resurgence or adaptability in this specific species.
Lines 65–74: This statement is the same as the previous line. Consider narrowing the focus to L. bostrychophila. Also, it would be a good idea to add how the strength of its endosymbionts specifically contributes to its success as a pest.
Lines.75–86: In my understanding, establishing a fact or adding a small paragraph about why previous approaches, such as PCR, were insufficient for understanding endosymbiont diversity and function.
Lines 106–124: Describe why this specific site was selected? Was it the pest infestation level, accessibility, or representativeness of other grain storage facilities? Providing this information will surely improve the rationale for the study.
Lines 125–137 (DNA Extraction): As a reader in the scientific community, it seemed interesting and shocking to see the use of a soil DNA kit for insect DNA extraction, as it could easily raise concerns about the suitability of the kit for this application. Can you explain and clarify the reason and rationale behind selecting this kit was chosen. Was it providing superior yield and compatibility with microbial DNA in insects? Also, have authors considered alternative DNA extraction kits were considered or tested for comparative efficiency.
Lines 183–199 (Establishing Rickettsia Abundance System): Can you clarify whether these temperature regimes (28°C, 35°C, and 37°C) mimic natural conditions or are primarily experimental. Also, discuss whether other environmental factors (e.g., humidity) were controlled to isolate temperature as the variable.
Lines 263–270: Performing ANOVA is appropriate, but it would be more helpful to specify whether assumptions for this test, such as normality, homogeneity of variance, and power, were checked. If not, consider discussing corrective measures, such as data transformation or alternative non-parametric tests.
Lines 276–284: The values related to Rickettsia's dominance are impressive and intriguing. If possible, can you also Include a brief comparison with similar studies on other stored-product pests? Is this level of dominance typical, or does it suggest a unique symbiotic relationship in L. bostrychophila?
Lines 314–324: This result presented in the study about the role of Rickettsia in host fitness is interesting. However, in my understanding, the section also lacks discussion of whether this reduction in lifespan aligns with findings from other studies on symbiont-host interactions. Also, could you please highlight whether this fitness reduction is due to direct physiological impacts or secondary effects like reduced reproduction?
Lines 335–342: The survival curves are insightful, but consider providing a hypothesis on why Rickettsia abundance has a more pronounced effect in later developmental stages. Could this be linked to cumulative metabolic stress or a critical dependence on Rickettsia-produced metabolites during adulthood?
Lines 373–387: Complexity related to the microbiome has always been a challenge, and I can clearly see that this diversity in the microbiome is not fully discussed in this manuscript. Can you explain with suitable references how the presence of other microorganisms potentially interacts with Rickettsia? Do they get competition inhibition or symbiotic type relation? Could these interactions influence host fitness or symbiont abundance?
Lines 402–411: Provide previous literature or references that can be used to elaborate on whether the reduced egg production is likely caused by a lack of essential nutrients or other physiological impacts mediated by Rickettsia. Additionally, consider discussing whether this reduction has implications for pest management strategies, such as targeting symbionts to suppress pest populations.
Lines 428–435: In my understanding, it would be good to include more specific examples about how geographic variation might influence Rickettsia abundance or its effects on host fitness. What about the impact of environmental factors on combined effects such as temperature, humidity, and moisture or genetic diversity of L. bostrychophila?
Lines 437–439: Could you provide some examples of how the system could be optimized. For instance, would testing additional environmental factors such as diet composition or alternative temperature ranges yield more comprehensive insights into the symbiont-host dynamics?
Author Response
Comments 1
Lines 43–46: Can authors provide some values for the economic losses caused by L. bostrychophila globally or regionally? This data would be invaluable in highlighting the pest's importance to the general audience. Additionally, given the authors' expertise, could they provide some examples of stored products, such as what kind of food or medicinal herbs, to further enrich the manuscript?
Response:Thank you for your suggestion. According to your suggestion, we have made the following modifications in the context.
Liposcelis bostrychophila Badonnel (Psocoptera: Liposcelididae) is a widely distributed pest that infests stored products, including food(Wheat, rice, coffee, etc.), medicinal herbs(such as Corchorus capsularis ), books, archives, and biological specimens [1,2,3]. Weight loss of 9.7% of broken wheat kernels due to damaged by L. bostrychophila for three months [4]. The annual losses caused by booklice in commercial rice storage in India (40,000 tons) and Indonesia (150,000 tons) were estimated at £115,000 ($180,000) and £50,000 ($75,000), respectively [5].
Comments 2
Lines 47–49: Could the authors kindly expand on the nature of pathogenic microorganisms and allergenic proteins carried by L. bostrychophila? I understand that the authors' knowledge in this area is extensive, and I believe that providing specific diseases or allergens would further strengthen the argument.
Response:Thank you for your valuable comments. Indeed, Liposcelis bostrychophila (L. bostrychophila) is not only a serious stored product pest but also may carry pathogenic microorganisms and contain allergenic proteins, which may pose a threat to human health. Liposcelis bostrychophila may carry pathogenic microorganisms such as Rickettsia felis (. Mediannikov, O. et al., J. Infect. Dis.,2022). Some studies have shown that the Rickettsia felis strains in Liposcelis bostrychophila can cause fever and experimental pneumonia in mammals. In addition, the allergenic proteins in Liposcelis bostrychophila are also considered as potential factors for triggering allergic reactions. Some studies have indicated that the Lip b1 protein in Liposcelis bostrychophila may be a novel allergenic protein( Ishibashi, O. et al., Allergy. 2017 ). This protein may cause symptoms such as human respiratory tract allergies, and the presence of this protein undoubtedly increases the health risks of the population. I will further expand this part according to your suggestions to further strengthen my argument. Thank you again for your feedback. Your suggestions are of great help to me.
Comments 3
Lines 50–52: it would be a good idea to mention why current control measures fail or their limitations in addressing these traits. This will provide a seamless transition into the study's significance.
Response:Thank you for your valuable comments. Due to the fact that Liposcelis bostrychophila has the characteristics of asexual reproduction, strong mobility, and high adaptability, if the control measures are insufficient, its population will quickly recur, which makes effective pest management a major challenge [8, 9]. However, the existing control strategies, such as chemical insecticides, usually cannot fully cope with these characteristics. The widespread use of chemical insecticides may lead to the development of resistance, and the strong mobility of this pest makes it difficult to control its outbreaks. In addition, the asexual reproduction of Liposcelis bostrychophila enables its population to recover rapidly even after chemical treatment. These limitations highlight the necessity of seeking more innovative and sustainable control methods. Therefore, understanding the factors that affect the population growth of Liposcelis bostrychophila is crucial for the development of more effective pest management strategies.
Comments 4
.Lines 54–64: It is a well-established fact that there is a link between endosymbionts and population growth. It is important, but if the authors can provide a brief explanation of why this is relevant to L. bostrychophila, it would be a great addition. It would also be great if the authors could explain how endosymbionts influence pest resurgence or adaptability in this specific species.
Response:Thank you for your valuable suggestions. The reasons you mentioned regarding the failure of existing control measures are indeed an important issue worthy of in-depth discussion. In the thesis, I have already mentioned that Liposcelis bostrychophila has characteristics such as asexual reproduction, strong mobility, and high adaptability, which lead to the easy recurrence of its population. However, I did not discuss in detail the limitations of current control measures when dealing with these characteristics. According to your suggestions, I will further supplement in the thesis: the challenges faced by current control measures (such as chemical insecticides), including the development of resistance, and the rapid population recovery caused by asexual reproduction and other issues. These factors make it difficult for existing control methods to completely solve the problem of the population growth of Liposcelis bostrychophila . Therefore, it emphasizes the need for innovative and sustainable control methods. Thank you again for your feedback. I will further improve the content of the thesis according to your suggestions to make it clearer and more complete.
Comments 5
Lines 65–74: This statement is the same as the previous line. Consider narrowing the focus to L. bostrychophila. Also, it would be a good idea to add how the strength of its endosymbionts specifically contributes to its success as a pest.
Response:Thank you for your valuable suggestions. Since previous studies have reported inconsistent results regarding the types of endosymbiotic bacteria in Liposcelis bostrychophila , and there is no clear understanding of how the symbiotic bacteria within Liposcelis bostrychophila specifically affect its biological characteristics, we intended to initiate our research on Liposcelis bostrychophila by referring to the studies on the interactions between other pests and their symbiotic bacteria. That's why we didn't solely focus on Liposcelis bostrychophila at the beginning. Indeed, enhancing the intensity of its endosymbionts is of great significance for studying the interaction between Liposcelis bostrychophila and its endosymbionts. However, due to the tiny size of Liposcelis bostrychophila , it is extremely difficult to artificially increase the abundance of endosymbionts within its body.
Comments 6
Lines.75–86: In my understanding, establishing a fact or adding a small paragraph about why previous approaches, such as PCR, were insufficient for understanding endosymbiont diversity and function.
Response:Thank you for your valuable suggestions. I understand what you mean. You suggested that in the paper, we should further clarify why previous methods (such as PCR) have deficiencies in understanding the diversity and functions of symbiotic microorganisms. I will add a short paragraph according to your suggestion to clearly point out the limitations of the PCR method. For example, with this addition, I hope to more clearly explain why the metagenomics method provides a more comprehensive and efficient solution in this regard. Thank you again for your suggestions, and I will strengthen the elaboration of this part during the revision process.
Comments 7
Lines 106–124: Describe why this specific site was selected? Was it the pest infestation level, accessibility, or representativeness of other grain storage facilities? Providing this information will surely improve the rationale for the study.
Response:Thank you for your valuable suggestions. In our preliminary research, we collected samples of Liposcelis bostrychophila from more than 20 locations. Through preliminary data analysis, we found that the population of Liposcelis bostrychophila in Hubei region grew the fastest. Therefore, we selected the Hubei strain as the key research object of this study. The population characteristics of this region provide relatively prominent representativeness and research value for our study. In addition, the populations of Liposcelis bostrychophila in other regions are also continuously tracked and sampled in subsequent research, with the aim of conducting a comparative analysis of the population differences in different regions.
Comments 8
Lines 125–137 (DNA Extraction): As a reader in the scientific community, it seemed interesting and shocking to see the use of a soil DNA kit for insect DNA extraction, as it could easily raise concerns about the suitability of the kit for this application. Can you explain and clarify the reason and rationale behind selecting this kit was chosen. Was it providing superior yield and compatibility with microbial DNA in insects? Also, have authors considered alternative DNA extraction kits were considered or tested for comparative efficiency.
Response:Your concern is of great significance. As you mentioned before, we took into consideration that since we were analyzing the DNA of microorganisms, whether it was from the soil or from inside insects. Meanwhile, when extracting the DNA of symbiotic microorganisms within insects, it was necessary to grind the insect bodies. In this way, the interior of the insects could be fully exposed, which is similar to the situation in the soil environment. Therefore, we made such an attempt. Judging from the results of metagenomics, the extracted DNA meets the requirements for metagenomic sequencing. That's why we didn't consider making a comparison between the two extraction methods.
|
Table 1 Metagenomic sequencing data statistics of L. bostrychophila endosymbiotic bacteria |
|||||
|
sample |
reads |
Number of bases/bp |
Average sequence length/bp |
Q30/% |
GC/% |
|
HBSJ_1 |
57697870 |
8624689740 |
149 |
90.83 |
35.53 |
|
HBSJ_2 |
74874710 |
11192946916 |
149 |
87.27 |
35.81 |
|
HBSJ_3 |
90286622 |
13494705670 |
149 |
88.12 |
35.56 |
Comments 9
Lines 183–199 (Establishing Rickettsia Abundance System): Can you clarify whether these temperature regimes (28°C, 35°C, and 37°C) mimic natural conditions or are primarily experimental. Also, discuss whether other environmental factors (e.g., humidity) were controlled to isolate temperature as the variable.
Response:Thank you for your valuable suggestions. Regarding the temperature conditions (28°C, 35°C, and 37°C), these temperature settings are mainly based on the needs of the experimental design, rather than a complete reproduction of natural conditions. We selected these temperatures to simulate the different environmental temperatures that might be encountered within insects, in order to evaluate the changes in the abundance of Rickettsia at these different temperatures. Among them, 28°C is the suitable temperature for Liposcelis bostrychophila , 37°C is referred to from previous studies, and 35°C is a randomly selected intermediate value. As for other environmental factors, we did control the humidity in the experiment to ensure that temperature was the only variable factor. The control of humidity was to exclude its potential impact on the abundance of Rickettsia, so that we could more accurately analyze the impact of temperature on the abundance of microorganisms.
Lines 263–270: Performing ANOVA is appropriate, but it would be more helpful to specify whether assumptions for this test, such as normality, homogeneity of variance, and power, were checked. If not, consider discussing corrective measures, such as data transformation or alternative non-parametric tests.
Response:Thank you for your valuable comments. In our study, we indeed used the SPSS 27 software to conduct the Analysis of Variance (ANOVA) and selected the Duncan's method for the post-hoc test. Before performing the ANOVA, we had carried out hypothesis testing on the data to ensure the validity and reliability of the analysis.
Comments 10
Lines 276–284: The values related to Rickettsia's dominance are impressive and intriguing. If possible, can you also Include a brief comparison with similar studies on other stored-product pests? Is this level of dominance typical, or does it suggest a unique symbiotic relationship in L. bostrychophila?
Response:Thank you for your valuable comments. Regarding the dominant position of Rickettsia in Liposcelis bostrychophila (L. bostrychophila), indeed, as you said, this result is impressive and thought-provoking. According to our current research, the abundance of Rickettsia in this species is significantly higher than that of other microorganisms, indicating that it may occupy an important symbiotic position in L. bostrychophila. We have also reviewed the studies on Rickettsia or similar symbiotic microorganisms in other stored product pests. In the existing literature, although Rickettsia is common in a variety of insects, its dominant position and abundance vary greatly among different species. For example, in Rhyzopertha dominica and Sitophilus oryzae, Rickettsia is not part of the main symbiotic flora and has a low abundance. However, in L. bostrychophila studied by us, the high abundance of Rickettsia may imply a unique symbiotic relationship between this insect and the microorganism. Of course, future research still needs to further explore the mechanism of this symbiotic relationship and whether it has similar manifestations in other stored product pests.
Comments 11
Lines 314–324: This result presented in the study about the role of Rickettsia in host fitness is interesting. However, in my understanding, the section also lacks discussion of whether this reduction in lifespan aligns with findings from other studies on symbiont-host interactions. Also, could you please highlight whether this fitness reduction is due to direct physiological impacts or secondary effects like reduced reproduction?
Response:Thank you for your valuable comments. Regarding the impact of Rickettsia on the adaptability of the host, especially the finding that the host's lifespan is shortened in our study, we have indeed observed a decrease in the lifespan of the host. We have noticed that this result is consistent with some studies in the existing literature. For example, in the research on the influence of the endosymbiotic bacterium Rickettsia on the biological characteristics of the sweetpotato whitefly (Bemisia tabaci), the infection of Rickettsia can prolong the lifespan of adult sweetpotato whiteflies. As for whether this shortening of lifespan is related to changes in reproductive ability, at present, we cannot directly answer whether there is a relationship between the two. Thank you for your attention. In subsequent research, we will take this aspect into consideration. In addition, I will further expand the discussion of this part in the thesis and supplement it with a comparative analysis of relevant literature to enhance the reliability and explanatory power of the results.
Comments 12
Lines 335–342: The survival curves are insightful, but consider providing a hypothesis on why Rickettsia abundance has a more pronounced effect in later developmental stages. Could this be linked to cumulative metabolic stress or a critical dependence on Rickettsia-produced metabolites during adulthood?
Response:Thank you for your valuable comments. Regarding the issue that the abundance of Rickettsia has a more significant impact on the later development stages in the survival curve, what you proposed is indeed a direction worthy of further in-depth exploration. We are aware that this impact may be related to the host's metabolic requirements, physiological changes during the development stages, or changes in the dependence on the metabolites produced by Rickettsia. However, since our current research is relatively preliminary and we have not fully explored the potential mechanisms of this phenomenon, in subsequent research, we plan to conduct an in-depth analysis of this issue and explore the possible mechanisms.
Comments 13
Lines 373–387: Complexity related to the microbiome has always been a challenge, and I can clearly see that this diversity in the microbiome is not fully discussed in this manuscript. Can you explain with suitable references how the presence of other microorganisms potentially interacts with Rickettsia? Do they get competition inhibition or symbiotic type relation? Could these interactions influence host fitness or symbiont abundance?
Response:Thank you for your valuable comments. Regarding the diversity of the microbial community, we indeed realize that the discussion of this part is insufficient in the current manuscript. Our research mainly focuses on the dominant endosymbiotic bacteria in Liposcelis bostrychophila . As for the overall diversity of the microbial community of Liposcelis bostrychophila , it has been explored in another one of our articles. For the relevant research content, please refer to: Zhang Kaizhi, Duan Yiwen, Guo Ziqiang, et al. Metagenome-based analysis of the diversity and functions of symbiotic microorganisms in the stored grain pest Liposcelis bostrychophila [J]. Journal of the Chinese Cereals and Oils Association, 2023, 38(12): 46-52. DOI: 10.20048/j.cnki.issn.1003-0174.000569. The complexity of the microbial community is indeed an important challenge in the research, especially in terms of host-microbe interactions. Currently, there is no relevant literature on the research of the interactions between Rickettsia and other microorganisms in Liposcelis bostrychophila , so we cannot give a definite answer in this regard. However, we understand the importance of this issue and plan to further explore the potential interactions between Rickettsia and other microorganisms in future research, especially in aspects such as competition, inhibition, or symbiotic relationships within the microbial community. Thank you for your feedback. We will pay attention to this complexity in subsequent research and hope to reveal the potential impacts of these microbial interactions on host fitness and the abundance of Rickettsia through more detailed experimental designs.
Comments 14
Lines 402–411: Provide previous literature or references that can be used to elaborate on whether the reduced egg production is likely caused by a lack of essential nutrients or other physiological impacts mediated by Rickettsia. Additionally, consider discussing whether this reduction has implications for pest management strategies, such as targeting symbionts to suppress pest populations.
Response:Thank you for your suggestions. Currently, no one has published research on the mechanism of the impact of Rickettsia on Liposcelis bostrychophila . However, some studies have shown that Rickettsia can affect the nutrition and defense of the sweetpotato whitefly (Bemisia tabaci), which in turn leads to an increase in the egg-laying amount of the sweetpotato whitefly. Yes, pest management targeting Rickettsia and other symbiotic bacteria is a promising strategy. By interfering with the relationship between Rickettsia and the host, it may effectively reduce the reproductive rate, thereby suppressing the pest population and enhancing the effectiveness of pest control. In the article "Hype or opportunity? Using microbial symbionts in novel strategies for insect pest control", the use of symbionts to suppress pest populations is discussed in detail. Regarding the impact of Rickettsia on egg-laying amount, some existing studies have indicated that Rickettsia, by influencing the physiological mechanisms of the host, especially in terms of nutritional metabolism and the immune system, may lead to a decrease in the host's reproductive ability. For example, Rickettsia may affect the development of the ovaries and egg production by altering the amino acid or fatty acid metabolism of the host, resulting in nutritional deficiencies (Moran & Yun, 2015; Dillon & Dillon, 2004). In addition, Rickettsia may affect the reproductive ability by suppressing the host's immune system or competing with other microorganisms (Herren & McMahon, 2017). We will further explore the specific impact of Rickettsia on the host's egg production in future research and consider the potential significance of this impact on pest management strategies.
Comments 15
Lines 428–435: In my understanding, it would be good to include more specific examples about how geographic variation might influence Rickettsia abundance or its effects on host fitness. What about the impact of environmental factors on combined effects such as temperature, humidity, and moisture or genetic diversity of L. bostrychophila?
Response:Thank you for your suggestion. The abundance of endosymbiotic bacteria in Liposcelis bostrychophila is influenced by various factors. It is also affected by geographical populations. However, it remains unclear how these factors exert their influences.
Comments 16
Lines 437–439: Could you provide some examples of how the system could be optimized. For instance, would testing additional environmental factors such as diet composition or alternative temperature ranges yield more comprehensive insights into the symbiont-host dynamics?
Response:Thank you for your suggestion. Yes, further optimizing the temperature range of the research materials and the research materials themselves can provide a more comprehensive understanding of the symbiosis-host dynamics.
Reviewer 2 Report
Comments and Suggestions for Authors
This paper deals with an intriguing interaction between an economically relevant insect and its main bacterial symbiont Rickettsia. Most of the results presented here may be useful for the scientific community dealing with the management of L. bostrychophila; however, several data are apparently missing and they should be better discussed. Some specific comments are listed below:
At the end of the Introduction, a short paragraph indicating what are the scientific questions addressed by this manuscript would be needed. At this step, it is still not clear if the authors aim to focus on specific bacterial taxa (i.e. reproductive manipulators that have been listed above) or to describe the overall microbiome diversity of L. bostrychophila.
Materials and methods
L118-124. The explanation of crawler tower is not totally clear to me. What do mean when saying that it was made to obtain clean insect: was it meant to make the insect crawl on a clean surface?
L200 “Quantification of Rickettsia” instead of “standard curve”
L209-210. Are those primers a newly designed pair? please specify
L258-260: Please provide information about how data where collected (e.g. what do you mean for developmental duration: days per instar, total days from egg to adult, …; how did you measure survival rates. no. of live specimens reaching adulthood, no. of live stages per instar, ….; what parameter did you measure to assess reproductive period or reproductive ability? and so on).
Results
3.1. Dominant Symbiotic Bacterial Species in L. bostrychophila: Many important data are missing from this part (es. tot. no. of species that you have retrieved, abundance as the higher levels, information about non-bacterial associates). Moreover, a section describing the results of gene prediction is totally missing. All of this would justify why you applied a metagenomics approach instead of a targeted NGS sequencing (e.g. bacterial V4)
L279 “51 genera were identified at the genus level”. at the genus level is redundant, please delete.
Figure 1 is very hard to read, please consider to increase its size.
Standard curve section: Despite interesting as a proof of the reliability of your test, I don’t think that it would be needed to specify those data in the manuscript, as this is not a paper specifically designed to present and validate a qPCR method. Please consider to move this part into the supplementary material.
L297: Diverse Rickettsia Abundance System paragraph: I believe it would be worth mentioning here that you did not find Rickettsia being completely eliminated at any temperature.
Table 3: please check carefully the column naming: only starting from Longevity a unit of measurement is indicated. Moreover, as the table should be easy to understand even as a stand-alone file, I suggest to explain the abbreviations (e.g. d=days I presume).
Figure 4 (and related text): did you normalize the Rickettsia copy number taking into account the total DNA load. I presume that adult samples may have a higher total DNA density than eggs, and this may hamper the comparability among samples.
3.4. Life Table Parameters: Please inform about any possible changes in the sex ratio. Did you find males at any temperature? This may be suggestive of parthenogenetic induction. I believe it would be relevant also maentioning if males are never seen.
Figure 5. I suggest to replace this graphical representation with a Kaplan-meier graph, which would be much clearer. Kaplan-meier would also be the most correct way to represent and compare data statistically as well.
Discussion
L389, 395, and below. Please replace Chinese characters using the Latin alphabet.
General comment on discussion. I suggest to include a short paragraph about possible roles of Rickettsia as a reproductive manipulator, although your aims were not to specifically assess this type of interaction. Moreover, if you have data from the metagenomics study that suggest any other functional role it should be included here.
Author Response
Comments 1
Materials and methods
L118-124. The explanation of crawler tower is not totally clear to me. What do mean when saying that it was made to obtain clean insect: was it meant to make the insect crawl on a clean surface?
Response:Thank you for the valuable comments from the judges. The "crawler tower" mentioned in the article is designed to obtain Liposcelis bostrychophila with clean epidermis and free from feed as much as possible. The purpose of this device is to allow Liposcelis bostrychophila to crawl on a clean surface, so as to remove the attached feed and impurities, thus ensuring that relatively clean insects can be obtained.
Comments 2
L200 “Quantification of Rickettsia” instead of “standard curve”
Response:Thank you for the judges' suggestions. We understand the opinions of the judges and agree to use the expression "Quantification of Rickettsia" to describe our experimental method more accurately. We will revise the relevant parts in the article and use more appropriate terms to ensure that the description of the research method is clearer and more accurate.
Comments 3
L209-210. Are those primers a newly designed pair? please specify
Response:Thank you for your question. Yes, these primers are a pair newly designed by us. During the design process, we took into account factors such as the length of the primers, GC content, and annealing temperature. Through the melting curve analysis of quantitative fluorescence PCR, we verified the specificity of the primers. The results showed that they have good specificity and reliability when amplifying the target sequence.
Comments 4
L258-260: Please provide information about how data where collected (e.g. what do you mean for developmental duration: days per instar, total days from egg to adult, …; how did you measure survival rates. no. of live specimens reaching adulthood, no. of live stages per instar, ….; what parameter did you measure to assess reproductive period or reproductive ability? and so on).
Response:Thank you for your question. During the data collection and analysis process, we referred to the following literature for our operations. For the specific methods, you can refer to the detailed descriptions in this literature. Therefore, we did not elaborate on it in this study, and all the data collection methods followed the standard operating procedures described in this literature. Chi, H.; Güncan, A.; Kavousi, A.; Gharakhani, G.; Atlihan, R.; Özgökçe, M.S.; Shirazi, J.; Amir-Maafi, M.; Maroufpoor, M.; Taghizadeh, R. TWOSEX-MSChart: the key tool for life table research and education. Entomol. Gen. 2022, 42, 845–849.
Chi, H.; You, M.; Atlıhan, R.; Smith, C. L.; Kavousi, A.; Özgökçe, M. S.; Güncan, A.; Tuan, S.-J.; Fu, J.-W.; Xu, Y.-Y.; Zheng, F.-Q.; Ye, B.-H.; Chu, D.; Yu, Y.; Gharekhani, G.; Saska, P.; Gotoh, T.; Schneider, M. I.; Bussaman, P.; Gökçe, A.; Liu, T.-X. Age-stage, two-sex life table: an introduction to theory, data analysis, and application. Entomol. Gen. 2020, 40, 103–124. Available online: https://api.semanticscholar.org/CorpusID:216280814.
comments 5
Results
3.1. Dominant Symbiotic Bacterial Species in L. bostrychophila: Many important data are missing from this part (es. tot. no. of species that you have retrieved, abundance as the higher levels, information about non-bacterial associates). Moreover, a section describing the results of gene prediction is totally missing. All of this would justify why you applied a metagenomics approach instead of a targeted NGS sequencing (e.g. bacterial V4)
Response:Thank you for your valuable comments. Regarding the diversity of the microbial community, we indeed realize that the discussion of this part is insufficient in the current manuscript. In this study, we mainly focused on the dominant endosymbiotic bacteria in Liposcelis bostrychophila, so we did not conduct an in-depth discussion on the diversity of its overall microbial community. However, we have already discussed the overall diversity of the microbial community of Liposcelis bostrychophila in detail in another article. For the relevant research content, please refer to: Zhang Kaizhi, Duan Yiwen, Guo Ziqiang, et al. Metagenome-based analysis of the diversity and functions of symbiotic microorganisms in the stored grain pest Liposcelis bostrychophila [J]. Journal of the Chinese Cereals and Oils Association, 2023, 38(12): 46-52. DOI: 10.20048/j.cnki.issn.1003-0174.000569.
L279 “51 genera were identified at the genus level”. at the genus level is redundant, please delete.
Response:Thank you for your valuable suggestions. Regarding the expression "genera were identified at the genus level", indeed, as you pointed out, "at the genus level" is redundant. We will delete this part in the revised version to make the expression more concise. Thank you again for your meticulous review and suggestions, and we will make the revisions according to your comments.
Figure 1 is very hard to read, please consider to increase its size.
Response:Thank you for your valuable comments. Regarding the readability issue of Figure 1, we understand your concerns. To improve the clarity of the chart, we can provide an enlarged version of the chart and attach it to the supplementary materials. This can ensure that readers can view the content of the chart more clearly. Thank you again for your meticulous review, and we will make corresponding adjustments according to your suggestions.
Standard curve section: Despite interesting as a proof of the reliability of your test, I don’t think that it would be needed to specify those data in the manuscript, as this is not a paper specifically designed to present and validate a qPCR method. Please consider to move this part into the supplementary material.
Response:Thank you for your valuable comments. We have taken note of the suggestions you made regarding the standard curve section. We understand that the focus of this article is not specifically on presenting and validating the qPCR method, so the data of the standard curve may not need to be listed in detail in the main text. We will, as per your suggestion, move this part of the content to the supplementary materials to maintain the conciseness of the article and ensure that readers can access the relevant data as needed. Thank you again for your review and suggestions, and we will make revisions based on your feedback.
L297: Diverse Rickettsia Abundance System paragraph: I believe it would be worth mentioning here that you did not find Rickettsia being completely eliminated at any temperature.
Response:Thank you for your valuable comments. The suggestions you put forward regarding the section on the Rickettsia abundance system are of great value. Indeed, we did not observe the phenomenon of Rickettsia completely disappearing at any temperature. We will add this point in the revised version. Thank you again for your meticulous review. We will make the necessary modifications according to your suggestions to make the article more perfect.
Table 3: please check carefully the column naming: only starting from Longevity a unit of measurement is indicated. Moreover, as the table should be easy to understand even as a stand-alone file, I suggest to explain the abbreviations (e.g. d=days I presume).
Response:Thank you for your valuable comments. Regarding the issue of unit labeling in the column names of the table, we will make adjustments in the revised version to ensure that the unit is indicated for each column, so as to improve the clarity and readability of the table. In addition, for the abbreviations in the table, we will add corresponding explanations. For example, we will explain that "d" stands for "day" to help readers better understand the content of the table. Thank you for your suggestions on the readability of the table. We will make modifications according to your feedback to make it more understandable. Thank you again for your meticulous review and suggestions.
Figure 4 (and related text): did you normalize the Rickettsia copy number taking into account the total DNA load. I presume that adult samples may have a higher total DNA density than eggs, and this may hamper the comparability among samples.
Response:Thank you for your question. Regarding the standardization of the copy number of Rickettsia, I would like to add that we used the absolute quantitative fluorescence PCR method to detect the copy number of Rickettsia in Liposcelis bostrychophila. In the calculation process, we divided the total copy number of Rickettsia in the sample by the number of Liposcelis bostrychophila in the sample, thus obtaining the copy number of Rickettsia in each Liposcelis bostrychophila. The advantage of this method is that it avoids the influence of the direct total DNA load on the copy number result, because we standardize it according to the number of individual Liposcelis bostrychophila, rather than relying solely on the total DNA load. Therefore, even if the total DNA density of the adult samples is higher than that of the egg samples, we can still ensure the comparability of the copy number data.
3.4. Life Table Parameters: Please inform about any possible changes in the sex ratio. Did you find males at any temperature? This may be suggestive of parthenogenetic induction. I believe it would be relevant also maentioning if males are never seen.
Response:Thank you for your questions and suggestions. Regarding the change in the sex ratio, we did observe the distribution of sexes at different temperatures. However, we did not observe the appearance of male individuals at any of the experimental temperatures. Since male individuals did not appear at all in the experiment, we did not conduct further analysis on the change in the sex ratio. If there is new data in the future or under different experimental conditions, we will continue to pay attention to the possibility of male individuals and further explore the possibility of parthenogenesis. Thank you for your meticulous review.
Figure 5. I suggest to replace this graphical representation with a Kaplan-meier graph, which would be much clearer. Kaplan-meier would also be the most correct way to represent and compare data statistically as well.
Response:Thank you for your valuable suggestions. We also believe that the Kaplan-Meier curve is an effective survival analysis method and can display the data very well. However, in this study, we used the Sex Life Table software, which can not only analyze the survival rate but also analyze the reproductive situation simultaneously. In order to maintain the consistency of the analysis method, we chose to use this software to generate data and create charts.
Comments 6
Discussion
L389, 395, and below. Please replace Chinese characters using the Latin alphabet.
Response:Thank you for your suggestion. We will replace all Chinese characters in the text with Latin letters according to your request to ensure compliance with the journal's formatting requirements. The corresponding changes will be made when the revised version is submitted. Thank you again for your meticulous review and valuable comments.
General comment on discussion. I suggest to include a short paragraph about possible roles of Rickettsia as a reproductive manipulator, although your aims were not to specifically assess this type of interaction. Moreover, if you have data from the metagenomics study that suggest any other functional role it should be included here.
Response:Thank you for your valuable suggestions. We highly agree with your opinion that the potential of Rickettsia as a reproductive manipulation factor is indeed an interesting research direction. Although the main objective of this study is not specifically to evaluate this type of interaction, we will add a brief paragraph in the discussion section to explore the potential role of Rickettsia in reproductive manipulation. Regarding the metagenomic study, although we did not find other functional roles of Rickettsia in the data, we did not conduct an in-depth discussion on this part. Thank you again for your meticulous review and constructive suggestions.
Reviewer 3 Report
Comments and Suggestions for Authors
Comments for authors
Title
I suggest little modification in title “Abundance of the Dominant Endosymbiont Rickettsia and Fitness of the Stored-Product Pest Liposcelis bostrychophila (Psocoptera: Liposcelididae)”.
Summary
Line: line 14: Replace population with a more suitable word.
Line 16: Endosymbionts play a significant regulatory role in the population dynamics. Of what? The sentence is incomplete. Please rewrite it for better understanding.
Line 17: Rickettsia was the predominant genus of symbiotic microorganisms. Present in? Please check this sentence also for better understanding.
Abstract
Line 22: Please rewrite as “Endosymbiotic bacteria are key factors which regulate biological traits of Liposcelis bostrychophila”.
Line 26: and low levels of these bacteria abundance. Remove word abundance or rewrite it to avoid grammatical error.
Line 30: of the total. Please write as “of the total share”.
Line 37-38: prolonging egg hatching time. Prolonging egg hatching will enhance vulnerability of eggs to biological control agents or other control tactics. How it can be beneficial for the pest?
Please add brief future prospects of this study at the end of abstract section.
Introduction
Line 45: stored products. Please mention names of some products.
Line 46: hides in its habitat. Please write a line about its habitat.
Line 68-69: Add more information in this sentence for better clarity.
Line 71: to studying. Please write “to study”
Line 81: Yusuf [31]. Its Yusuf et al. [31]
Line 97: Perotti [36]. Its Perotti et al. [36]. Please check all citations for necessary corrections.
Cite this also in introduction “Comparative microbiome analysis of Diaphorina citri and its associated parasitoids Tamarixia radiata and Diaphorencyrtus aligarhensis reveals Wolbachia as a dominant endosymbiont”.
The last paragraph of introduction section is too lengthy. I would suggest that authors should split this in two smaller fragments. Moreover, objectives must be clearly mentioned at the end of this section which are currently missing.
Materials and Methods
2.1. Insect culture
108: L. bostrychophila specimens. Better to write stage of insect and write full form of insect at the start of sentence i.e. Liposcelis bostrychophila
Line 110: Also mention country name.
Line 113: Appropriate feed. The feed ingredients mentioned some lines below should be listed here at first appearance of word feed.
Section 2.2: line 129-130, Elaborate on how these samples differ from one another? or if they are biological replicates.
Section 2.2.3: Add details on whether specific metrics like GC content, duplication rates, or overall read quality were assessed to ensure data integrity.
Section 2.2.4: Clarify the alignment parameters used in Bowtie2 to ensure reproducibility and interpretability of the results.
Line 148-149: Specify the criteria used to determine the success of purification, such as yield or removal of adapter dimers.
Section 2.2.6: Specify if additional databases or methods were considered for validation of species annotation results.
Line 270: 40. Please follow Journal guidelines for citations style. [40]
Results
Please carefully check the caption for table 5, also figure (5b), (6b), (7a) are mixed with another letter, please pay attention to minor details. The quality of figure could also be enhanced.
Discussion
The manuscript contains Chinese phrases in the discussion section. Are these a mistake, or do they represent important statements that cannot be translated or cited? Please clarify this in the manuscript.
Please carefully review the discussion section for any citation format inconsistencies with the rest of the paper.
The discussion is too general and lacks depth. The authors should strengthen it and consider organizing the content under different subheadings.
Author Response
Comments 1
I suggest little modification in title “Abundance of the Dominant Endosymbiont Rickettsia and Fitness of the Stored-Product Pest Liposcelis bostrychophila(Psocoptera: Liposcelididae)”.
Response:
Thank you for your suggestion. We fully agree with your proposed revision of the title. The minor adjustments you mentioned will help improve the clarity of the title, and we will make the changes based on your advice to ensure a more accurate representation of the core content of the study. Once again, we appreciate your thorough review and valuable feedback.
Comments 2
Line: line 14: Replace population with a more suitable word.
Response:Thank you for your suggestion. We understand that the use of "population" in the text may not be entirely accurate, and we will select a more appropriate term based on your feedback to ensure greater precision in the expression. Once again, we appreciate your careful review and valuable advice.
Line 16: Endosymbionts play a significant regulatory role in the population dynamics. Of what? The sentence is incomplete. Please rewrite it for better understanding.
Response:Thank you for your suggestion. The sentence you mentioned was indeed incomplete. In order to express it more clearly, we will revise the sentence to: "Endosymbionts play a significant regulatory role in the population dynamics of their host organisms." This revision makes the sentence more complete and better conveys the intended meaning. Once again, thank you for your thorough review and valuable feedback.
Line 17: Rickettsia was the predominant genus of symbiotic microorganisms. Present in? Please check this sentence also for better understanding.
Response:Thank you for your suggestion. The sentence you mentioned was indeed not complete enough, and we will make revisions to improve the clarity of the expression. The revised sentence could be: "Rickettsia was the predominant genus of symbiotic microorganisms, present in the Liposcelis bostrychophila." This revision makes the sentence more complete and provides clearer reference. Once again, thank you for your thorough review and valuable feedback.
Comments 3
Line 22: Please rewrite as “Endosymbiotic bacteria are key factors which regulate biological traits of Liposcelis bostrychophila”.
Response:Thank you for your suggestion. Based on your feedback, we have revised the sentence to: "Endosymbiotic bacteria are key factors that regulate the biological traits of Liposcelis bostrychophila." This revision makes the sentence more concise and clearer in expression. Once again, thank you for your thorough review and valuable advice.
Line 26: and low levels of these bacteria abundance. Remove word abundance or rewrite it to avoid grammatical error.
Response:Thank you for your suggestion. Based on your feedback, we will remove "abundance" or adjust the sentence to avoid grammatical errors. The sentence can be revised to: "and low levels of these bacteria." This revision makes the sentence more concise and grammatically correct. Once again, thank you for your thorough review and valuable advice.
Line 30: of the total. Please write as “of the total share”.
Response:Thank you for your suggestion. If you would like to use "of the total share," the sentence can be revised to: "of the total share." This modification better aligns with the expression you requested. Thank you again for your careful review and valuable feedback.
Line 37-38: prolonging egg hatching time. Prolonging egg hatching will enhance vulnerability of eggs to biological control agents or other control tactics. How it can be beneficial for the pest?
Response:Thank you for your question. Extending the egg incubation time may indeed increase the susceptibility of the eggs to biological control agents or other control methods. However, symbiotic bacteria have a dual role. They are both a burden to the host insect, as the host needs to provide energy and nutrients for their growth and development, and also beneficial to the host's growth, development, and reproduction. Therefore, although research indicates that Rickettsia delays the development time of Liposcelis bostrychophilaeggs, overall, Rickettsia is beneficial to the population development of Liposcelis bostrychophila.
Please add brief future prospects of this study at the end of abstract section.
Response:Thank you for your suggestion. According to your request, I will add a brief outlook on future research prospects at the end of the abstract.
Comments 4
Line 45: stored products. Please mention names of some products.
Response:Thank you for your suggestion. When mentioning "storage products" in line 45, I will add the names of some specific products.
Line 46: hides in its habitat. Please write a line about its habitat.
Response:hank you for your feedback. Based on your request, I will list some common habitats of this organism.
Line 68-69: Add more information in this sentence for better clarity.
Response:Thank you, judges, for your valuable comments. In line with your suggestions, I have added more specific information to the original text, elaborating in detail on the role of endosymbionts in the interaction between hosts and insects, as well as how they affect the ecological functions and adaptability of the hosts.
Line 71: to studying. Please write “to study”
Response:Thank you, judges, for your corrections. I've changed "to studying" to "to study" as per your suggestion, making the sentence more accurate and grammatically correct. Thanks for your meticulous feedback.
Line 81: Yusuf [31]. Its Yusuf et al. [31]
Response:Thank you for the comments from the judges. I have already modified "Yusuf [31]" to "Yusuf et al. [31]" according to your suggestion to ensure compliance with the correct format and convention of literature citation. Thank you for your meticulous correction.
Line 97: Perotti [36]. Its Perotti et al. [36]. Please check all citations for necessary corrections.
Response:Thank you for the judge's suggestions. I have already changed "Perotti [36]" to "Perotti et al. [36]" based on your feedback. Moreover, I have checked all the citations in the text to ensure they conform to the norms and made the necessary revisions. Thank you again for your meticulous review.
Cite this also in introduction “Comparative microbiome analysis of Diaphorina citri and its associated parasitoids Tamarixia radiata and Diaphorencyrtus aligarhensis reveals Wolbachia as a dominant endosymbiont”.
Response:Thank you for the judge's suggestions. I have added a citation for the article "Comparative microbiome analysis of Diaphorina citri and its associated parasitoids Tamarixia radiata and Diaphorencyrtus aligarhensis reveals Wolbachia as a dominant endosymbiont" in the introduction section and ensured that this literature is appropriately placed in the text. Thank you for your valuable comments and guidance.
The last paragraph of introduction section is too lengthy. I would suggest that authors should split this in two smaller fragments. Moreover, objectives must be clearly mentioned at the end of this section which are currently missing.
Response:Thank you for the valuable comments from the reviewer. I have split the last paragraph of the introduction section into two paragraphs to improve the readability and conciseness of the paragraphs. In addition, I have clearly listed the research objectives at the end of the introduction section to ensure that the content of this part is complete and clear. Thank you for your guidance and suggestions.
Comments 7
Materials and Methods
2.1. Insect culture
108:Liposcelis bostrychophilaspecimens. Better to write stage of insect and write full form of insect at the start of sentence i.e. Liposcelis bostrychophila
Response:Thank you for your suggestion. Indeed, it will be clearer to explicitly state the developmental stages of the insects and use the full name "Liposcelis bostrychophila" at the beginning of the sentence. I will make the modifications accordingly.
Line 110: Also mention country name.
Response:Thank you for your suggestion. I will also make sure to mention the name of the country.
Line 113: Appropriate feed. The feed ingredients mentioned some lines below should be listed here at first appearance of word feed.
Response:Thank you for the valuable suggestions from the judge. According to your comments, I will list all the relevant feed ingredients when the term "feed" is mentioned for the first time in the article to improve the accuracy and clarity of the article.
Section 2.2: line 129-130, Elaborate on how these samples differ from one another? or if they are biological replicates.
Response:Thank you for the valuable suggestions from the judge. Yes, these samples are biological replicates, which means they were independently sampled and processed under the same conditions with the aim of ensuring the reliability and consistency of the results.
Section 2.2.3: Add details on whether specific metrics like GC content, duplication rates, or overall read quality were assessed to ensure data integrity.
Response:Thank you for the valuable suggestions from the judge! Regarding data integrity, we have indeed evaluated several important indicators in the analysis, including GC content, repeatability rate, and overall read quality. We will include this part of the content in the supplementary materials.
Section 2.2.4: Clarify the alignment parameters used in Bowtie2 to ensure reproducibility and interpretability of the results.
Response:Thank you for the valuable suggestions from the judge. We have already made the necessary modifications.
Line 148-149: Specify the criteria used to determine the success of purification, such as yield or removal of adapter dimers.
Response:Thank you for the valuable suggestions from the judges. We have already made the necessary revisions.
Section 2.2.6: Specify if additional databases or methods were considered for validation of species annotation results.
Response:Thank you for the valuable suggestions from the judge! Yes, we used other databases to verify the results of species annotation. Since the focus of our research is on the dominant endosymbiotic bacteria of Liposcelis bostrychophilaand there is a limit to the length of the paper, we did not elaborate on this part in detail in the paper. Thank you for the judge's suggestion.
Line 270: 40. Please follow Journal guidelines for citations style. [40]
Response:Thank you for your suggestion. I will make the modifications accordingly.
Comments 8
Results
Please carefully check the caption for table 5, also figure (5b), (6b), (7a) are mixed with another letter, please pay attention to minor details. The quality of figure could also be enhanced.
Response:Thank you for the meticulous review by the judge. Regarding the title of Table 5, as well as Figures 5b, 6b, and 7a, we have carefully checked them and made the necessary revisions. Thank you for your valuable comments. We will continue to improve to enhance the quality of the thesis.
Comments 9
Discussion
The manuscript contains Chinese phrases in the discussion section. Are these a mistake, or do they represent important statements that cannot be translated or cited? Please clarify this in the manuscript.
Response:Thank you for pointing that out. Regarding the Chinese phrases that appeared in the discussion section, we have checked and confirmed that this might be an error caused by a typesetting issue. We have already corrected this part and ensured that all the content is in English. Thank you for your meticulous review.
Please carefully review the discussion section for any citation format inconsistencies with the rest of the paper.
Response:Thank you for pointing it out. We have already made corrections to this part.
The discussion is too general and lacks depth. The authors should strengthen it and consider organizing the content under different subheadings.
Response:Thank you for your valuable suggestions. We have noticed the problem that the discussion section is too general and lacks depth. Next, we will refine and expand the discussion section, increase the depth of the content, and introduce different subheadings as suggested to better organize the content, thus improving its readability and logicality. Thank you for your patient review and guidance.
Round 2
Reviewer 1 Report
Comments and Suggestions for Authors
Job well done. Thank you for addressing comments
Author Response
Comments 1: Job well done. Thank you for addressing comments.
Response 1: Thank you for your recognition and diligent efforts on behalf of our manuscript.
Reviewer 2 Report
Comments and Suggestions for Authors
The authors provided suitable rensponses to almost all of my previous comments; some suggestions to better incorporate the answers in the text are listed below:
PREVIOUS QUESTION: L118-124. The explanation of crawler tower is not totally clear to me. What do mean when saying that it was made to obtain clean insect: was it meant to make the insect crawl on a clean surface? RESPONSE: Thank you for the valuable comments from the judges. The "crawler tower" mentioned in the article is designed to obtain Liposcelis bostrychophila with clean epidermis and free from feed as much as possible. The purpose of this device is to allow Liposcelis bostrychophila to crawl on a clean surface, so as to remove the attached feed and impurities, thus ensuring that relatively clean insects can be obtained.
NEW SUGGESTION: According to your answer: please modify the sentence in L151-152: “ The eggs continued to be cultured under the same conditions for 30 days; after hatching nymphs were cleaned from any attached feed and impurities using a crawler tower”.
PREVIOUS QUESTION: L209-210. Are those primers a newly designed pair? please specify. RESPONSE: Thank you for your question. Yes, these primers are a pair newly designed by us. During the design process, we took into account factors such as the length of the primers, GC content, and annealing temperature. Through the melting curve analysis of quantitative fluorescence PCR, we verified the specificity of the primers. The results showed that they have good specificity and reliability when amplifying the target sequence.
NEW SUGGESTION: I agree with your decision to include the qPCR details in the supplementary material. Please specify that the reader should refer to the supplementary file when mentioning quantitative PCR analysis.
PREVIOUS QUESTION: 3.4. Life Table Parameters: Please inform about any possible changes in the sex ratio. Did you find males at any temperature? This may be suggestive of parthenogenetic induction. I believe it would be relevant also maentioning if males are never seen. Response:Thank you for your questions and suggestions. Regarding the change in the sex ratio, we did observe the distribution of sexes at different temperatures. However, we did not observe the appearance of male individuals at any of the experimental temperatures. Since male individuals did not appear at all in the experiment, we did not conduct further analysis on the change in the sex ratio. If there is new data in the future or under different experimental conditions, we will continue to pay attention to the possibility of male individuals and further explore the possibility of parthenogenesis. Thank you for your meticulous review.
NEW SUGGESTION: I believe it would be worth mentioning that at this point you did not record any change in parthenogenetic behaviour. You could add a sentence at the end of paragraph 3.4, for example: “The decrease of Ricketssia did not affect reproductive traits, as no males were recorded at any of any of the experimental temperatures” or something similar.
Author Response
Comments 1: PREVIOUS QUESTION: L118-124. The explanation of crawler tower is not totally clear to me. What do mean when saying that it was made to obtain clean insect: was it meant to make the insect crawl on a clean surface? RESPONSE: Thank you for the valuable comments from the judges. The "crawler tower" mentioned in the article is designed to obtain Liposcelis bostrychophila with clean epidermis and free from feed as much as possible. The purpose of this device is to allow Liposcelis bostrychophila to crawl on a clean surface, so as to remove the attached feed and impurities, thus ensuring that relatively clean insects can be obtained.
NEW SUGGESTION: According to your answer: please modify the sentence in L151-152: “ The eggs continued to be cultured under the same conditions for 30 days; after hatching nymphs were cleaned from any attached feed and impurities using a crawler tower”.
Response 1:Thank you for your suggestion. We have already made revisions to the text.
Comments 2: PREVIOUS QUESTION: L209-210. Are those primers a newly designed pair? please specify. RESPONSE: Thank you for your question. Yes, these primers are a pair newly designed by us. During the design process, we took into account factors such as the length of the primers, GC content, and annealing temperature. Through the melting curve analysis of quantitative fluorescence PCR, we verified the specificity of the primers. The results showed that they have good specificity and reliability when amplifying the target sequence.
NEW SUGGESTION: I agree with your decision to include the qPCR details in the supplementary material. Please specify that the reader should refer to the supplementary file when mentioning quantitative PCR analysis.
Response 2:Thank you for your valuable suggestions. In accordance with your comments, we have clearly indicated in the main text that readers should refer to the supplementary materials for detailed information on the quantitative PCR analysis. The specific revisions are reflected in the manuscript.
Comments 3: PREVIOUS QUESTION: 3.4. Life Table Parameters: Please inform about any possible changes in the sex ratio. Did you find males at any temperature? This may be suggestive of parthenogenetic induction. I believe it would be relevant also maentioning if males are never seen. Response:Thank you for your questions and suggestions. Regarding the change in the sex ratio, we did observe the distribution of sexes at different temperatures. However, we did not observe the appearance of male individuals at any of the experimental temperatures. Since male individuals did not appear at all in the experiment, we did not conduct further analysis on the change in the sex ratio. If there is new data in the future or under different experimental conditions, we will continue to pay attention to the possibility of male individuals and further explore the possibility of parthenogenesis. Thank you for your meticulous review.
NEW SUGGESTION: I believe it would be worth mentioning that at this point you did not record any change in parthenogenetic behaviour. You could add a sentence at the end of paragraph 3.4, for example: “The decrease of Ricketssia did not affect reproductive traits, as no males were recorded at any of any of the experimental temperatures” or something similar.
Response 3:Thank you for your valuable suggestions. In accordance with your comments, we have added a statement at the end of Section 3.4 to clarify that no changes related to parthenogenetic behavior were observed during the experiment. The specific revision is as follows: "No males were recorded at any of the experimental temperatures." Thank you for your meticulous review and guidance.
Reviewer 3 Report
Comments and Suggestions for Authors
Comments for authors
Please italicize technical terms in introduction.
Line 49: Please rewrite for better understanding.
Line 218: were obtained.
Line 102: Please also add citation of this. Ashraf HJ, Ramos Aguila LC, Akutse KS, Ilyas M, Abbasi A, Li X, Wang L. Comparative microbiome analysis of Diaphorina citri and its associated parasitoids Tamarixia radiata and Diaphorencyrtus aligarhensis reveals Wolbachia as a dominant endosymbiont. Environ Microbiol. 2022, 24:1638-1652.
Some tables are missing in revised version. Previous version has more tables. Please check.
Author Response
Comments 1: Please italicize technical terms in introduction.
Response 1:Thank you for your suggestion. We have already italicized all the professional terms in the introduction section.
Comments 2: Line 49: Please rewrite for better understanding.
Response 2:Thank you for your feedback. We have revised Line 49 of the introduction section.
Comments 3: Line 218: were obtained.
Response 3:Thank you for your correction. We have revised the expression in Line 218.
Comments 4: Line 102: Please also add citation of this. Ashraf HJ, Ramos Aguila LC, Akutse KS, Ilyas M, Abbasi A, Li X, Wang L. Comparative microbiome analysis of Diaphorina citri and its associated parasitoids Tamarixia radiata and Diaphorencyrtus aligarhensis reveals Wolbachia as a dominant endosymbiont. Environ Microbiol. 2022, 24:1638-1652.
Resposne 4:Thank you for your suggestion. We have added the literature citation you mentioned to Line 102.
Comments 5: Some tables are missing in revised version. Previous version has more tables. Please check.
Response 5:Thank you for your reminder. Regarding the issue of the missing tables you mentioned, due to the suggestions from other reviewers, the original section of the method "2.4. Standard Curve" has been moved to the supplementary materials section. Therefore, these tables are not shown in the revised version.